# MCCE: Missingness-aware Causal Concept Explainer

## Abstract

Causal concept effect estimation is gaining increasing interest in the field of interpretable machine learning. This general approach explains the behaviors of machine learning models by estimating the causal effect of human-understandable concepts, which represent high-level knowledge more comprehensibly than raw inputs like tokens. However, existing causal concept effect explanation methods assume complete observation of all concepts involved within the dataset, which can fail in practice due to incomplete annotations or missing concept data. We theoretically demonstrate that unobserved concepts can bias the estimation of the causal effects of observed concepts. To address this limitation, we introduce the Missingness-aware Causal Concept Explainer (MCCE), a novel framework specifically designed to estimate causal concept effects when not all concepts are observable. Our framework learns to account for residual bias resulting from missing concepts and utilizes a linear predictor to model the relationships between these concepts and the outputs of black-box machine learning models. It can offer explanations on both local and global levels. We conduct validations using a real-world dataset, demonstrating that MCCE achieves promising performance compared to state-of-the-art explanation methods in causal concept effect estimation.

## 1 Introduction

Machine learning models explained through concept-based methods are often more intuitive than those based solely on raw inputs like tokens or pixels (Poeta et al., 2023). Unlike traditional approaches that attribute model decisions to low-level features, such as individual pixels in an image or tokens in text, concept-based methods leverage high-level semantic knowledge derived from these inputs. These methods facilitate a deeper understanding of how models make decisions by aligning their internal representations with concepts that are comprehensible to humans. By focusing on high-level concepts, stakeholders can better assess the model's reasoning process. This is especially significant in areas like healthcare (Cutillo et al., 2020; Rasheed et al., 2022) and finance (Giudici & Raffinetti, 2023; Zhou et al., 2022), where trust and transparency are critical.

The gold standard for assessing a concept-based explanation is comparing its output to the causal effect of concepts (Wu et al., 2023). Causal effect estimation measures the direct impact of changing a specific concept on the outcome, while holding all other concepts constant. This approach goes beyond simple associations, which are prone to confounding effects, by identifying how altering a specific concept causally influences the model's predictions (Moraffah et al., 2020). However, current concept-based causal explanation methods usually assume that the entire set of involved concepts is completely observed in the dataset. In reality, the identification of concepts from data can vary between experts or automated systems, and one or many concepts may not be annotated in the entire dataset (Ghorbani et al., 2019). As a result, complete observation and annotation of all relevant concepts are not guaranteed in real-world applications, highlighting the need for methods that can handle incomplete or missing concept data.

In this paper, we conduct a mathematical analysis showing that the presence of unobserved concepts hinders the unbiased estimation of concepts' causal effects. To address this challenge, we propose a framework called Missingness-aware Causal Concept Explainer (MCCE). MCCE captures the impact of unobserved concepts by constructing pseudo-concepts that are orthogonal to observed

concepts. By modeling the relationship between concepts and a black-box model's output with a linear function, MCCE can not only estimate causal concept effects for individual samples (local explanation) and but also elucidate general rules used by a model to make decisions (global explanations). MCCE can also function as an interpretable prediction model if trained with groundtruth labels. The architecture of MCCE is depicted in Figure 1.

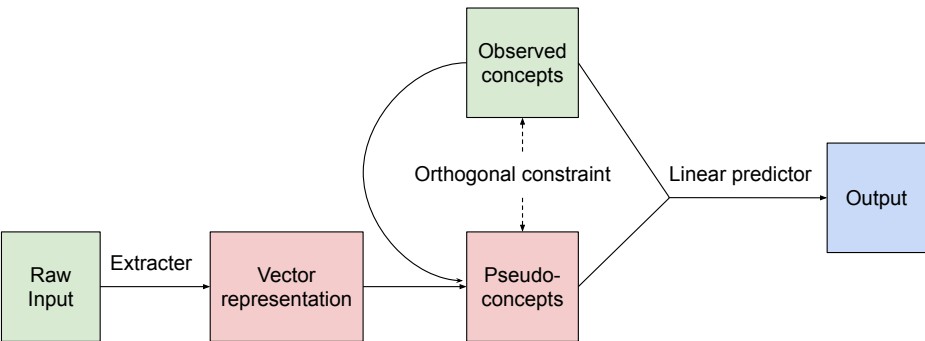

Figure 1: The architecture of MCCE. Given an input sample, a vector representation is extracted. Pseudo-concepts are constructed under the constraint that they are orthogonal to the observed concepts, ensuring that the pseudo-concepts offers information to compensate for any lost information from unobserved concepts. Then a linear predictor is trained on the concatenation of observed concepts and the pseudo-concepts to approximate the behaviors of a black-box model. The entire pipeline can be trained end-to-end.

We summarize our main contributions as follows:

- We demonstrate that violating the assumption of complete observation of concepts, which is commonly imposed in existing research, can lead to biased estimation of causal concept effects.
- We propose the Missingness-aware Causal Concept Explainer (MCCE), to our best knowledge, the first concept-based causal effect estimation framework that takes the existence of unobserved concepts into consideration.
- Empirical results show that our proposed MCCE achieves promising performance in estimating the Individual Concept Causal Effect Errors (ICaCE-Error) on a real-world dataset. Meanwhile, it can provide global interpretations of a model and can act as an interpretable white-box prediction model.

## 2 RELATED WORK

Explaining the behaviors of black-box machine learning models has been drawing interest from researchers in the past decade. Various methods have been proposed to estimate the contribution of input to models' output. Learned weights can be used to denote the importance of features (Olden & Jackson, 2002; Zhou et al., 2016; Molnar, 2020). Permutation-based methods evaluate feature importance by measuring how the model's prediction performance changes when the values of a single feature are randomly shuffled (Altmann et al., 2010; Lundberg, 2017; Smith et al., 2020). Gradient-based methods interpret machine learning models by analyzing the gradients of the model's output with respect to its input (Sundararajan et al., 2017; Kim et al., 2018; Srinivas & Fleuret, 2020). However, these approaches focus on individual input features instead of summarizing the effect of high-level semantic concepts.

The concept-based explanation uses high-level semantic concepts to interpret a black-box machine learning model's behaviors. Koh et al. (2020) introduced the Concept Bottleneck Model, which

predicts human-interpretable concepts as intermediate variables first and then uses these predicted concepts to make final predictions with an interpretable model such as a linear regression. Since then, variants of approaches have been developed to build concept-based interpretation frameworks, such as Concept Transformer (Rigotti et al., 2021), Post-hoc Concept Bottleneck Models (Yuksek-gonul et al., 2022), Concept Embedding Models (Zarlenga et al., 2022), Logic Explained Networks (Ciravegna et al., 2023), Probabilistic Concept Bottleneck Models (Kim et al., 2023), Enerby-based Concept Bottleneck Models (Xu et al., 2024), among others.

Recent years have witnessed the rising interests in causal concepts effect estimation. Feder et al. (2021) proposed CausalLM to estimate concept-based causal effects by learning counterfactual representations via adversarial tasks. Ravfogel et al. (2020) introduced Iterative Nullspace Projection to learn the causal effect of a concept, which removes a concept from a representation vector by iteratively training linear classifiers to predict the attribute and projecting it onto the null space. Abraham et al. (2022) not only built a human-validated concept-based dataset with counterfactuals called Causal Estimation-Based Benchmark (CEBaB) but also found that many popular explanation methods, including those described above, can fail to accurately estimate the causal effects of models on their developed dataset. Wu et al. (2023) developed the Causal Proxy Model (CPM) which mimics the counterfactual behaviors of a model by creating representations that allow for intervention, achieving state-of-the-art performance on the CEBaB dataset. However, their approach assumes all the involved concepts are observed during the development process. Our work provides a theoretical analysis of the impact of unobserved concepts and, motivated by this analysis, proposes a solution to reduce the resulting bias. We also compare the accuracy of causal effect estimation using our proposed method against existing approaches on the CEBaB dataset.

## 3 MISSINGNESS-AWARE CONCEPT-BASED CAUSAL EXPLAINER

In this section, we introduce the problem settings, analyze the impact of unobserved concepts on the concepts' causal effect estimation, and provide detailed descriptions of the proposed MCCE.

**Causal structure** Let $U$ be the exogenous variable, $C_{ob}$ be the observed concepts, $C_{un}$ be the unobserved concepts, $X$ be the input data fed to a black-box model $\mathcal{N}$, and $\mathcal{N}(X)$ be the output of the model. The exogenous variable $U$ represents the complete state of the world. The input data $X$ is generated from $U$ and mediated by concepts $C_{ob}$ and $C_{un}$. A black-box machine learning model $\mathcal{N}$ takes the input $X$ and makes output for a specific prediction task. Figure 2 shows the causal structure in a graph as an illustration.

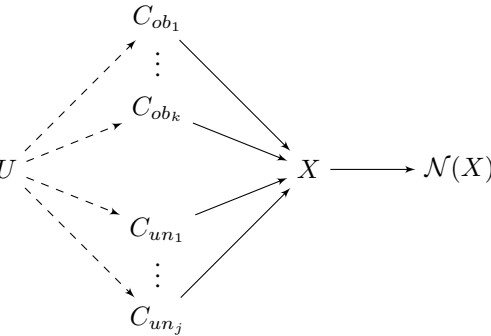

Figure 2: Causal structure graph. The impact of $U$ on $X$ is not only mediated by the observed concepts $C_{ob_1}, ... C_{ob_k}$ but also by the unobserved concepts $C_{un_1}, ... C_{un_j}$. In this work, we aim to account for the impact of unobserved concepts when estimating the causal effect of observed concepts, which has not been addressed in existing research. A backdoor path may exist from $C_{ob_1}$ to $\mathcal{N}(X)$, even though there is no direct path from $C_{ob_1}$ to $\mathcal{N}(X)$ and all other $C_{ob}$ are conditioned/blocked – this occurs through $U$ and one of $C_{un}$ (Pearl, 2009).

**Empirical Individual Concept Causal Effect ($\widehat{\text{ICaCE}}$)** Let $x^c$ denote an input sample with a concept value equal to $c$. For a black-box model $\mathcal{N}$, the empirical individual causal concept effect (Abraham et al., 2022) of changing the value of concept $C$ from $c$ to $c'$ on input $x$ is denied as

$$\widehat{\text{ICaCE}}_{\mathcal{N}}(x^{c \to c'}) = \mathcal{N}(x^{c \to c'}) - \mathcal{N}(x^c) \tag{1}$$

$\widehat{\text{ICaCE}}$ measures how perturbing a specific concept in a black-box model impacts the prediction of a specific input sample.

**ICaCE-Error** For a black-box model $\mathcal{N}$, a dataset $D$, and a distance metric Dist, the ICaCE-Error (Abraham et al., 2022) of an explanation method $\mathcal{E}$ for swapping the value of concept $C$ from $c$ to $c'$ is

$$\text{ICaCE-Error}_{\mathcal{N}}(\mathcal{E}) = \frac{1}{|D|} \sum_{x^c \in D} \text{Dist}\left(\widehat{\text{ICaCE}}_{\mathcal{N}}(x^c, x^{c \to c'}), \mathcal{E}(c, c'|x)\right) \tag{2}$$

ICaCE-Error measures the average distance between the $\widehat{\text{ICaCE}}$ and the estimation returned by explainer $\mathcal{E}$ across samples with concept value $c$. It is used as the quantitative evaluation metric for causal concept effect explanations (Abraham et al., 2022; Wu et al., 2023).

**Explaining causal concept effect with a linear model** We assume that there exists a linear explainer $\mathcal{E}^*$ for the **complete** concept set $C_{complete}$ that can perfectly explain the logit output of the $\mathcal{N}(X)$ with coefficients $\beta^*$:

$$\mathcal{E}^* = C_{complete}^T \beta^* = \mathcal{N}(X) \tag{3}$$

This linear assumption has been imposed by existing concept-based model explanation frameworks. Empirically, it can often hold approximately within concept-based explanation, as concepts tend to have proportional influences on the outcome across different scenarios. (Kim et al., 2018; Yuksekgonul et al., 2022; Tan et al., 2024).

Suppose we have an input sample $x$ with its $t$th concepts swapping from value $c$ to value $c'$. $\beta_t^*$ is the corresponding coefficient for the $t$th concept in Equation 3. Because all the remaining concepts are not changed, the $\widehat{\text{ICaCE}}$ can be rewritten as

$$\begin{aligned} \widehat{\text{ICaCE}}_{\mathcal{N}}(x^{c \to c'}) &= \mathcal{N}(x^{c \to c'}) - \mathcal{N}(x^c) \\ &= (c' - c)\beta_t^* \end{aligned} \tag{4}$$

For a linear explanation $\mathcal{E}$ with coefficients $\hat{\beta}$, the ICaCE-Error can be written as

$$\begin{aligned} \text{ICaCE-Error}_N(\mathcal{E}) &= \frac{1}{|D|} \sum_{x^c \in D} \text{Dist}\left(\widehat{\text{ICaCE}}_{\mathcal{N}}(x^c, x^{c \to c'}), \mathcal{E}(c, c'|x)\right) \\ &= \frac{1}{|D|} \sum_{x^c \in D} \text{Dist}\left(\beta_t^*(c' - c), \hat{\beta}_t(c' - c)\right) \\ &\propto \sum_{x^c \in D} \text{Dist}\left(\beta_t^*, \hat{\beta}_t\right) \end{aligned} \tag{5}$$

That being said, to minimize the ICaCE-Error, one needs to find unbiased estimators for the linear coefficients $\beta^*$. This is feasible with regular estimators such as an mean squared error (MSE) when all the involved concepts $C_{complete}$ are observed.

**Residual bias resulted from unobserved concepts** Complete observation is rarely available in real life. We write $C_{complete} = [C_{ob}, C_{un}]$ as the concatenation of observed concepts $C_{ob}$ and unobserved concepts $C_{un}$. We want to find $\hat{\beta}_{ob}$ for

$$\mathcal{N}(X) = C_{ob}^T \hat{\beta}_{ob} + C_{un}^T \hat{\beta}_{un} \tag{6}$$

where $\mathcal{N}(X) = C_{ob}^T \beta_{ob}^* + C_{un}^T \beta_{un}^*$.

The existence of residue $C_{un}^T \beta_{un}^* - C_{un}^T \hat{\beta}_{un}$ hinders the unbiased estimation of $\beta_{ob}^*$

**Missingness-aware Concept-based Causal Explainer (MCCE)**

The core idea behind MCCE is to compensate for information missed in observed concepts $C_{ob}$ by harnessing the raw input data. We create vectors, termed pseudo-concepts (denoted as $C_{pseud}$), that are orthogonal to the observed concept vectors using linear transformations from encoded input data. These pseudo-concepts are then combined with the observed concepts to train a linear model that approximates the output of a black-box model. The orthogonality of the pseudo-concepts to the observed concepts prevents collinearity, ensuring that these pseudo-concepts contribute information absent in the observed data.

Suppose $C_{ob}$ is a $n \times k$ vector and $C_{pseud}$ is a $n \times j$ vector, where $n$ is the sample size, $k$ is the number of observed concepts, and $j$ is a hyperparameter to denote the presumed number of pseudo-concepts. An extractor $\mathcal{M}(X)$ takes input $X$ and outputs a $n \times j$ dimensional vector $H$. We hypothesize that $H$ contains all necessary information about all concepts, including unobserved ones. To capture information from $H$ that is orthogonal to $C_{ob}$, we rewrite $C_{pseud}$ as below, inspired by recent work in factor analysis (Fan et al., 2024)

$$C_{pseud} = (I - P)H \tag{7}$$

where $I$ is the identification matrix and $P = C_{ob}(C_{ob}^T C_{ob})^{-1} C_{ob}^T$ is the orthogonal projection matrix onto the column space of $C_{ob}$. $(I - P)H$ denotes the residuals of $H$ after projecting onto the column space of $C_{ob}$. We have $C_{ob}^T C_{pseud} = 0$ because

$$
\begin{aligned}
C_{ob}^T C_{pseud} &= C_{ob}^T (I - P)H \\
&= (C_{ob}^T - C_{ob}^T P)H \\
&= (C_{ob}^T - C_{ob}^T)H \qquad \text{since } C_{ob}^T P = C_{ob}^T \text{ by defination} \\
&= 0
\end{aligned}
\tag{8}
$$

With $C_{ob}$ and $C_{pseud}$, we construct a linear predictor $\mathcal{N}(X)$:

$$\mathcal{N}(X) = C_{ob}^T \hat{\beta}_{ob} + C_{pseud}^T \hat{\beta}_{pseud} \tag{9}$$

We plug in Equation 7 to an MSE estimator to find $\hat{\beta}_{ob}$ such that

$$(\hat{\beta}_{ob}, \hat{\beta}_{pseud}) = \underset{\beta_{ob}, \beta_{pseud}}{\arg\max} \left( \frac{1}{2n} ||\mathcal{N}(X) - C_{ob}^T \beta_{ob} - (I - P)H\beta_{pseud}||_2^2 \right) \tag{10}$$

As a summary, our proposed MCCE converts input to a numerical vector and transforms this vector into pseudo-concepts that are orthogonal to the observed concepts. Then a linear predictor is used to approximate a black-box model's output using the observed concepts and the pseudo-concepts. Let $C_{x,ob}$ denote the observed concept vector of input $x$, the linear predictor $\mathcal{G}$ can be written as:

$$\mathcal{G}(C_{x,ob}, x, \mathcal{M}) = C_{x,ob}^T \hat{\beta}_{ob} + \left[ \left( I - C_{x,ob}(C_{x,ob}^T C_{x,ob})^{-1} C_{x,ob}^T \right) \mathcal{M}(x) \right]^T \hat{\beta}_{pseud} \tag{11}$$

To estimate the causal effect of a concept swapping from $c$ to $c'$ in the inference stage, we intervene the corresponding values in $C_{ob}$, with $x$ remaining unchanged. With $C_{x,ob}^{c \to c'}$ denoting swapping one of input $x$'s concepts from $c$ to $c'$, the MCCE explainer $\mathcal{E}_{MCCE}$ can be written as:

$$\mathcal{E}_{MCCE}(c, c'|x) = \mathcal{G}_{\mathcal{M}}(C_{x,ob}^{c, \to c'}, x) - \mathcal{N}(x) \tag{12}$$

## 4 EXPERIMENTS

### 4.1 DATASET

We use the CEBaB dataset Abraham et al. (2022) to validate our proposed MCCE. CEBaB contains restaurant reviews from OpenTable for sentiment analysis. Each text in CEBaB received a 5-star sentiment score from crowd workers and was annotated on four concept levels-ambiance, food, noise, and service, with the labels negative, unknown, and positive. It started with 2,299 original reviews and was expanded to 15,089 texts through modifications by human annotators. These annotators edited the reviews to reflect specific interventions such as changing food evaluations from positive to negative. As far as we know, it is the only dataset with human-verified approximate counterfactual text. The resulting dataset is divided into training, development, and testing partitions. The development and test sets serve to evaluate explanation methods. The ICaCE-Error of MCCE and the baselines are validated using the test set.

### 4.2 MCCE CONSTRUCTION

We finetune three different types of publicly available models for the multiclass semantic classification tasks of the CEBaB dataset: the base BERT (Devlin, 2018), the base RoBERTa (Liu, 2019), and Llama-3 (Dubey et al., 2024). These three models are different generations of transformer-based language models designed to capture contextual relationships within text and have been widely used in a wide range of NLP tasks. We use the last hidden states of the *cls* token for BERT and RoBERTa as the $H$ vector in Equation 7, and for Llama-3, we use the last hidden states of the last token. To explore how unobserved concepts influence the ICaCE-Error across different explainers, we omit each of the four attributes individually during the construction of MCCE and the baseline models. Additionally, we exclude every possible pair of concepts from these four attributes.

### 4.3 BASELINES

Let $x^c$ denote an input sample with a concept value $c$. We implement below methods as baselines to estimate the ICaCE-Error of swapping a concept value from $c$ to $c'$ for a black-box model $\mathcal{N}$.

**Approximate Counterfactuals**  As a baseline, we sample a factual input with the same concept labels as the $x_{sampled}^{c'}$ and use it as an approximate counterfactual. This approximate counterfactuals explainer can be formally written as

$$\mathcal{E}_{approx}(c, c'|x) = \mathcal{N}(x_{sampled}^{c'}) - \mathcal{N}(x^c) \tag{13}$$

**S-Learner**  The S-Learner, originally proposed for Conditional Average Treatment Effect (CATE) estimation (Künzel et al., 2019), is one of the top performers in the original CEBaB paper. It uses all the observed concepts to fit a logistic regression model $\mathcal{R}$ to predict the output of $\mathcal{N}$. At the inference stage, it compute the difference between the counterfactual concept vector $C_{x,ob}^{c \to c'}$ and the factual concept vector $C_{x,ob}^c$

$$\mathcal{E}_{S-Learner}(c, c'|x) = \mathcal{R}(C_{x,ob}^{c \to c'}) - \mathcal{N}(x^c) \tag{14}$$

**Input-based Causal Proxy Model**  The input-based Causal Proxy Model (CPM) (Wu et al., 2023) outperforms existing approaches on the CEBaB dataset. Given a counterfactual pair $(x^{c \to c'}, x^c)$, where $x^{c \to c'}$ is the human-created counterfactual sample, CPM concatenates a learnable token $tk_{c \to c'}$ to the end of the original text $x^c$ and train a language model $\mathcal{P}$, which shares the same architecture as the black-box model $\mathcal{N}$, to approximate the output of $\mathcal{N}$ with the counterfactual input by minimizing the smoothed cross-entropy (Hinton, 2015) as:

$$\mathcal{L}_{CPM} = CE\left(\mathcal{N}(x^{c \to c'}), \mathcal{P}(x^c, tk_{c \to c'})\right) \tag{15}$$

During the inference stage, the CPM measures the causal concept effect of swapping $c$ to $c'$ as the difference between $\mathcal{P}(x^c, tk_{c \to c'})$ and the black-box model's factual output $\mathcal{N}(x^c)$. The CPM explainer $\mathcal{E}_{CPM}$ can be written as

$$\mathcal{E}_{CPM}(c, c'|x) = \mathcal{P}(x^c, tk_{c \to c'}) - \mathcal{N}(x^c) \tag{16}$$

## 5 RESULTS

We conduct validation of MCCE alongside baseline methods using the CEBaB dataset. Table 1 reports the means and standard deviations of the ICaCE-Error when two concepts or one concept are unobserved. For a specific number of unobserved concepts, the means and standard deviations of ICaCE-Error are displayed for all possible combinations of unobserved concepts. To assess the ICaCE-Error, we employ L2, Cosine, and Norm distance metrics. MCCE outperforms S-Learner over all the metrics with either one or two concepts being unobserved. As S-Learner only trains a learner predictor with observed concepts, it can be recognized as a special case of MCCE that removed the components of the pseudo-concepts. The contrast between MCCE and S-Learner demonstrates that the capture of pseudo-concepts effectively mitigates the residue bias caused by the unobserved concepts. MCCE consistently achieves performance on par with or superior to the CPM across all considered distance metrics. When two out of the four concepts are unobserved, MCCE demonstrates a distinct advantage over the baselines in terms of Cosine distance, which prioritizes the directional alignment between vector pairs rather than merely their magnitude. While CPM accurately estimates causal concept effects by learning the impact of altering a concept value while keeping other elements constant, its performance declines with two concepts are unobserved compared to only one, particularly measured using Cosine distance. On the other hand, MCCE demonstrates robust performance especially when evaluated using Cosine distance, underscoring its effectiveness in mitigating residual bias to estimate the direction of causal concept effect through the construction of pseudo-concepts that are orthogonal to observed concepts.

Table 1: Means and standard deviations of ICaCE-Error (the lower, the better) of MCCE and baselines with three different types of models when two or one concepts are unobserved in the CEBaB dataset. Best results are bolded. For each number of unobserved concepts, means and standard deviations for all possible combinations of unobserved concepts are displayed.

| Model | Metric | *Two concepts are unobserved* | | | | *One concepts are unobserved* | | | |
| | | Approx | S-Learner | CPM | MCCE (ours) | Approx | S-Learner | CPM | MCCE (ours) |
|---|---|---|---|---|---|---|---|---|---|
| BERT | L2 | 1.52 | 1.29 | **1.01** | **1.02** | 1.52 | 1.10 | **0.98** | **0.99** |
| | | (0.06) | (0.05) | **(0.05)** | **(0.05)** | (0.05) | (0.02) | **(0.04)** | **(0.03)** |
| | Cosine | 0.75 | 0.64 | 0.60 | **0.56** | 0.75 | 0.60 | **0.56** | **0.57** |
| | | (0.03) | (0.03) | (0.03) | **(0.02)** | (0.03) | (0.03) | **(0.02)** | **(0.03)** |
| | Norm | 0.88 | 0.78 | **0.65** | **0.66** | 0.88 | 0.72 | **0.64** | 0.68 |
| | | (0.06) | (0.05) | **(0.04)** | **(0.04)** | (0.06) | (0.05) | **(0.05)** | (0.04) |
| RoBERTa | L2 | 1.48 | 1.30 | **0.99** | **0.98** | 1.48 | 1.15 | **0.99** | **0.95** |
| | | (0.06) | (0.04) | **(0.05)** | **(0.04)** | (0.06) | (0.03) | **(0.04)** | **(0.04)** |
| | Cosine | 0.72 | 0.65 | 0.61 | **0.57** | 0.72 | 0.60 | **0.57** | **0.57** |
| | | (0.04) | (0.04) | (0.03) | **(0.03)** | (0.04) | (0.04) | **(0.03)** | **(0.02)** |
| | Norm | 0.91 | 0.81 | **0.64** | **0.65** | 0.91 | 0.73 | **0.63** | 0.66 |
| | | (0.05) | (0.05) | **(0.04)** | **(0.05)** | (0.05) | (0.05) | **(0.05)** | (0.04) |
| Llama 3 | L2 | 1.32 | 0.95 | **0.81** | **0.82** | 1.32 | 0.90 | **0.81** | **0.78** |
| | | (0.05) | (0.03) | **(0.03)** | **(0.04)** | (0.05) | (0.03) | **(0.04)** | **(0.04)** |
| | Cosine | 0.64 | 0.55 | 0.55 | **0.50** | 0.72 | 0.60 | 0.57 | **0.51** |
| | | (0.03) | (0.03) | (0.02) | **(0.03)** | (0.04) | (0.04) | (0.03) | **(0.02)** |
| | Norm | 0.85 | 0.76 | 0.58 | **0.52** | 0.85 | 0.70 | 0.55 | **0.52** |
| | | (0.03) | (0.05) | (0.02) | **(0.05)** | (0.03) | (0.03) | (0.05) | **(0.02)** |

MCCE can offer a global interpretation of the impact each observed concept has on the output of a black-box model. This is demonstrated in Figure 3, which shows the coefficients for the attributes "Ambiance," "Service," and "Noise" in a five-class sentiment classification task. To adhere to the identification constraints required for multiclass classification, we designate the coefficients of the

1-star category as the baseline for comparison. For "Noise" and "Service" with positive attitudes, the coefficient shows increasing positive influence as the ratings increase, becoming most prominent at a 4-star rating. A positive "Ambiance" has a peaking impact at the 5-star rating instead of the 5-star rating. The negative attitudes toward the three attributes result in heavier negative influences on higher ratings. The "*unknown*" category has moderate impacts with smaller magnitudes of coefficients and tends to slightly lean towards mid-class like 3-star ratings. This figure demonstrates MCCE's ability to depict how each attribute's impact varies across different ratings for a given model, which is lacking in the CPM model.

MCCE not only provides causal explanations but also can function as an interpretable predictor. Table 2 displays MCCE's performance, measured by macro-F1 score, when it is used to directly learn sentimental outcomes instead of interpreting a black-box model's output. MCCE predictor achieves comparable performance when leveraging BERT and RoBERTa's hidden states compared to their black-box model counterpart. When leveraging an LLM as the extractor, MCEE only slightly underperforms compared to the Llama-3 black-box model. Notably, the performance of MCCE remains robust, showing only a marginal decrease, when two concepts are unobserved compared to just one. This again demonstrates that the pseudo-concepts can capture critical information that is missed from observed concepts.

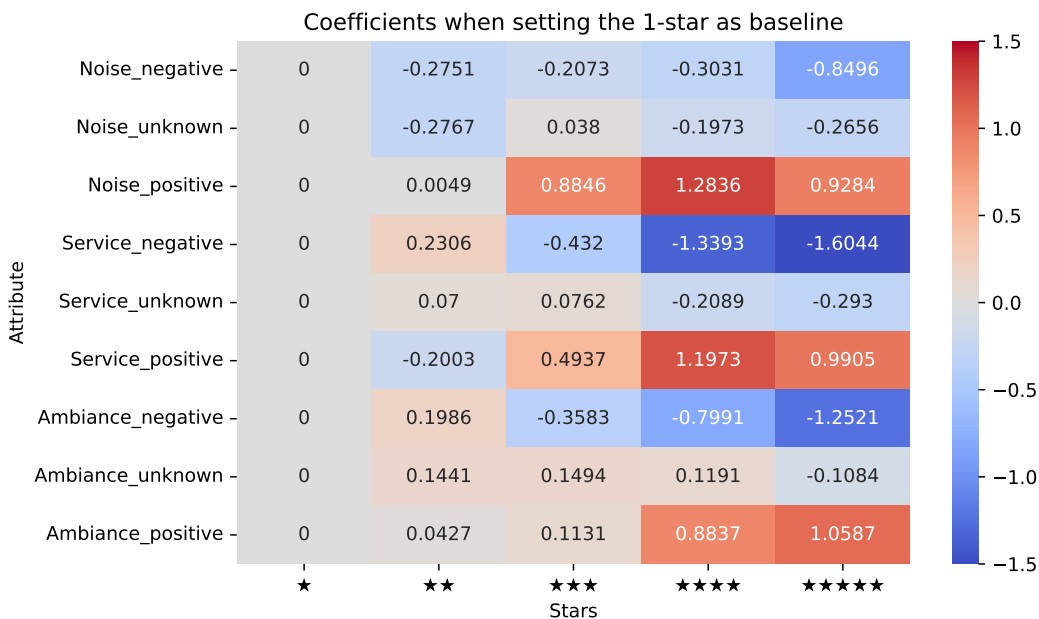

Figure 3: An illustration of the MCCE's global interpretation on a BERT model when the concepts of "Ambiance", "Service", and "Noise" are observed.

Table 2: Macro-F1 (the larger, the better) performance of MCCE on directly predicting the CEBaB outcomes. "**1 unobserved**" and "**2 unobserved**" indicate one and two concepts are unobserved. Means and standard deviations for all possible combinations of unobserved concepts are displayed.

| Model | Blackbox | MCCE (1 unobserved) | MCCE (2 unobserved) |
|---|---|---|---|
| BERT | 0.72 (0.02) | 0.72 (0.02) | 0.71 (0.02) |
| RoBERTa | 0.71 (0.02) | 0.72 (0.02) | 0.70 (0.03) |
| Llama-3 | 0.78 (0.01) | 0.75 (0.02) | 0.74 (0.02) |

## 6 DISCUSSION

In this work, we address a critical but previously unexplored question: How can we effectively measure the causal effect of concepts when some are unobserved? We introduce the Missingness-aware Concept-based Causal Explainer (MCCE), the first framework specifically designed to estimate the causal effects of concepts while accounting for the impact of those that are unobserved. MCCE innovatively constructs pseudo-concepts that are column-wise orthogonal to the observed concepts, enriching the model with complementary information that captures the influence of the missing concepts. Our experimental results on the CEBaB dataset demonstrate that MCCE achieves superior or, at the very least, comparable performance to existing baseline methods in scenarios where unobserved concepts are present.

Among the baseline methods, the CPM approach is the only one that matches MCCE's performance on some of the metrics. However, CPM methods depend on labeled counterfactual data for training, which may limit their practical applicability. In contrast, MCCE effectively utilizes only factual training data and does not require counterfactual data. Additionally, the coefficients derived from MCCE's linear predictor offer a direct, global interpretation of black-box models, a character absent in CPM.

The number of pseudo-concepts in MCCE is a hyperparameter that needs to be pre-selected. Empirically, we observe that a number of pseudo-concepts comparable to or slightly greater than the number of observed concepts tends to yield the best results. For example, in a scenario where one of four attributes in CEBaB is unobserved, with the remaining three attributes encoded into nine concepts (*"negative"*, *"unknown"*, *"positive"* for each attribute). MCCE shows optimal performance with nine or twelve pseudo-concepts. The theoretical rationale for the choice of the number of pseudo-concepts remains a subject for further investigation.

One limitation of our work is that MCCE is validated on one dataset. The CEBaB dataset, which, while comprehensive, contains only a limited number of labeled concepts. Though the construction of the orthogonal pseudo-concepts and the linear predictor allows MCCE to cope with a large number of concepts through straightforward modifications according to Fan et al. (2024), further empirical validation is necessary to fully establish its effectiveness across varied datasets. In addition, MCCE is designed to be modal-agnostic but needs further validation on modalities beyond text. Unfortunately, to our knowledge, CEBaB is the sole publicly available dataset that includes the labeled counterfactual data necessary for assessing causal concept effects. Further validation will be available only when more benchmark datasets for causal concept effect estimation are available.

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
