# OpenReview forum: "MCCE: Missingness-aware Causal Concept Explainer"
_ICLR.cc/2025/Conference — ICLR 2025 Conference Withdrawn Submission_

### Official Review · Reviewer_6434 · 2024-11-02

**Soundness:** 2
**Presentation:** 3
**Contribution:** 1
**Rating:** 3
**Confidence:** 4

**Summary:**

This work proposes Missingness-aware Causal Concept Explainer (MCCE), a concept-based causal explainability framework that aims to estimate the causal effect high-level concepts may have on a model’s output when the identity of all concepts is unknown. MCCE approaches this by considering both the effect of concepts whose labels are known and the potential effects of “discovered” concept directions that are orthogonal to those of concepts provided as labels. This work shows that MCCE can lead to results comparable to previous approaches, such as CPM, while enabling higher levels of interpretability and flexibility.

**Strengths:**

Thank you so much for submitting this work. I enjoyed reading this paper and appreciate the time taken to make it clear and easy to follow. Below are what I believe are this paper’s main strengths:

1. **[Significance] (Major)** The paper's main intent and goal, that of modeling causal effects from both observed and unobserved concepts, is important for key tasks in interpretability, causality, and XAI. As such, I believe this paper’s goals and studies can interest several research communities. Moreover, the results provide early signals that this approach may indeed work in practice and be easier to use than other existing approaches.
2. **[Quality] (Major)** The method is very well explained and elaborated. In particular, the authors did an amazing job of carefully motivating their approach.
3. **[Originality] (Minor)** I find the simplicity of modeling unobserved concepts as orthogonal directions to the set of observed concepts both interesting and useful. However, as I believe this has been explored in some previous works in a related but not exactly the same way, I mark this strength as only a minor strength.
4. **[Clarity] (Minor)** The paper is very well-written and easy to follow. I also really appreciate that the work is transparent in the way it discusses some of its limitations in its discussion.

**Weaknesses:**

In contrast, I believe the following are some of this work’s limitations:

1. **[Significance] (Critical)** The empirical evaluation is very limited compared to what one would expect from an ICLR paper. This is perfectly ok if the work instead focuses on providing novel theoretical results (e.g., proofs) or insights/unexpected realizations. However, I believe this work in its current state does not provide enough evidence or results (either theoretically or empirically) to entirely convince me that this represents a significant and well-defended contribution. For example, the proposed method is only evaluated on a single dataset (with only four concept groups), and even here, it is only superficially explored (there is a lot of emphasis on using different architectures and metrics, yet with the exception of the qualitative analysis of the coefficients in Figure 3, there is little analysis on things that go beyond analyzing simple performance). Moreover, the claimed theoretical contribution (e.g., see abstract) falls short of being a complete/well-detailed proof and is not formalized. See below for specific questions related to this matter.
2. **[Originality and Significance] (Critical)** At a high level, it appears that the method’s novelty relies heavily on the previous work by Abraham et al. (e.g., most of the definitions, baselines, and the dataset used come from Abraham et al.). Again, this is okay if there is a significant improvement over the work of Abraham et al. or if key unknown insights were provided that were not provided in that work. However, this doesn’t seem to be the case as the performance improvements over CPM are very marginal in some instances (and potentially not significant, see below). Moreover, the ideas explored here seem very related to those seen in previous works in concept-based XAI, such as completeness-aware concept discovery [CCD], and it would be extremely helpful to compare and contrast MCCE with something like CCD (at the very least, I believe this paper should be cited).
3. **[Quality] (Major)** The performance results in Table 1 seem to show that MCCE is only marginally better than CPM in some instances (e.g., the gains do not seem to fall outside their 95% confidence intervals) and worse in others. As such, it is difficult to build a cogent case for using MCCE over CPM, as the claim that MCCE needs pairs of counterfactuals for training can be overcome by constructing counterfactuals/approximations of counterfactuals when concept labels are available in a training set (see below for specific questions). If the claim is that MCCE is better than CPM in many instances, then I would suggest backing this up with proper statistical tests, as this conclusion does not seem to follow immediately when considering the variances in Table 1.
4. **[Quality] (Major)** This submission provides almost no details that would enable one to recreate the results observed in this paper. Several key hyperparameters (learning rates, epochs, model selection searching spaces, batch sizes, etc.) are never discussed, and no code was provided either. This goes against good scientific practices and hampers reproducing this work’s key results.
5. **[Quality] (Minor)** MCCE still requires one to determine how many unobserved concepts should be extracted. These unobserved concepts have not been studied at all, and no insights/ablations have been provided to suggest how one should select this value. There is a short discussion of this in Section 6, but no evidence or ablations are provided to back up the claims made in that section.

## References

- [CCD] Yeh, Chih-Kuan, et al. "On completeness-aware concept-based explanations in deep neural networks." *Advances in neural information processing systems* 33 (2020): 20554-20565.

**Questions:**

Currently, I am leaning towards rejecting this paper given its limited evaluation and my concerns regarding its originality vis-a-vis previous works. I am absolutely happy to be convinced that some or all of my conclusions are wrong and believe the following questions could help clarify/question some of my concerns:

1. **(Critical)** Do you have evidence of MCCE’s performance on other datasets? MCCE does not seem to require counterfactual pairs for training, and therefore, I do not believe that is a strong reason not to include further datasets as part of the evaluation. Even if that is the case, one can argue that these pairs could be constructed/approximated from data if we know some of the ground truth concepts during training (by looking at samples whose concept annotations differ in at most one concept as done by, say, semi-supervised disentanglement learning approaches). Finally, I can’t see a case for not being able to build even synthetic datasets that have all the required components for evaluation from commonly used benchmarks like [dSprites] or [3D Shapes]. Unfortunately, I strongly believe that, as the theoretical contributions of this paper are a bit limited, without any further evaluation, it is difficult to make a case for claiming that this work is backed up by enough evidence.
2. **(Critical)** One of the claimed contributions is that this work demonstrates “that violating the assumptions of complete observation of concepts … can lead to biased estimation of causal concept effects.” I understand the gist of this claim vis-a-vis what is discussed in Section 3. Yet, I would argue that this is not really a proof or a “demonstration.” Could you please formalize this proof, clearly stating its assumptions and its conclusions in a formal manner so that it is easier to understand its consequences? Moreover, if this is to be a contribution to this work, could you please elaborate on how it relates to previous theoretical results in causality that work with unobserved variables?
3. **(Critical)** How does this work compare to previous methods exploring unobserved concepts as orthogonal directions such as CCD (a quite popular approach in concept-based XAI)? I understand CCD is not designed with causality in mind (although it could easily be used to explore the same questions as done in this paper), yet if one trivially adds supervision to one or more of its discovered concepts, the others become the equivalent of orthogonal directions that may represent unobserved concepts. Therefore, I am interested in trying to understand how the key ideas of this work differ from those used by CCD (which I understand were done in a different context).
4. **(Critical)** Can one claim that the improvements of MCCE seen in Table 1 are significant with respect to the performance of CPM?
5. **(Critical)** What are the hyperparameters and training details for the results shown in Section 5? How was model selection performed? Was it fair for all baselines, or was the focus of hyper-optimization primarily on MCCE? It would strengthen this paper to include all key details as part of an appendix at the very least (as it is usually good practice).
6. **(Major)** Do you know if some of the unobserved concepts end up indeed aligning or matching with concepts that are known during evaluation but not provided during training? Having a better understanding of this can be critical to understanding whether this model properly uses its residual capacity to learn things that are indeed interpretable/useful.
7. **(Major)** An “explanation method” $\mathcal{E}$ is used from line 168 onwards without any definition of what this even means (the terms “explanation method” carry a lot of different meanings depending on the context). Could you please clarify what this means and what is the definition of this term?
8. **(Major)** Do you have any empirical evidence to back up the claims made on how the number of unobserved parameters is selected in Section 6?

### Minor Suggestions and Typos

Whilst reading this work, I found the following potential minor issues/typos which may be helpful when preparing a new version of this manuscript:

1. **(Potential Typo)** In Figure 1, “extracter” should probably be “extractor”
2. **(Potential Typo)** In line 113, “Enerby-based” should probably be “Energy-based”
3. **(Potential Typo)** In line 161, “denied as” should probably be “defined as”
4. **(Potential Typo)** In line 188, “its $t$th concepts” should probably be “its $t$-th concept”
5. **(Potential Typo)** In equation (8), “by defination” should probably be “by definition”
6. **(Potential Typo)** In line 308, “it compute the difference” should probably be “it computes the difference”
7. **(Nitpicking)** In lines 32-33 it is claimed that “Machine learning models explained through concept-based methods are often more intuitive than those based solely on raw inputs like tokens or pixel.” However, machine learning models are as intuitive regardless of the explanation used. What changes is the “intuitiveness” of the explanation, not of the underlying model.

## References

- [CCD] Yeh, Chih-Kuan, et al. "On completeness-aware concept-based explanations in deep neural networks." *Advances in neural information processing systems* 33 (2020): 20554-20565.
- [dSprites] Matthey, Loic, et al. "dsprites: Disentanglement testing sprites dataset." May 2017.
- [3D Shapes] Burgess, Chris, and Hyunjik Kim. "3d shapes dataset." 2018.

---

### Official Review · Reviewer_4axU · 2024-11-03

**Soundness:** 3
**Presentation:** 2
**Contribution:** 3
**Rating:** 3
**Confidence:** 3

**Summary:**

This paper proposes a Missingness-aware Causal Concept Explainer (MCCE), a novel framework to estimate causal concept effects when not all concepts are observable by learning to amount for bias.

**Strengths:**

1.The problem that causal concepts are missing is practically important and worth a study.

**Weaknesses:**

1. Theoretical analysis should be strengthened,
2. The empirical justifications are not well presented.

**Questions:**

1. I think the main theoretical contribution, the claim “We theoretically demonstrate that unobserved concepts can bias the estimation of the causal effects of observed concepts” is trivial. Is there any justification that under which scenario these “missing” ones are estimable by tour method? Especially from observational data?

2. The core model design relies on a linear predictor to estimate the bias. This claim “Our framework learns to account for residual bias resulting from missing concepts and utilizes a linear predictor to model the relationships between these concepts and the outputs of black-box machine learning models.” seems to be a main structure to deal with missing ones. I still have puzzle why a linear predictor is “enough” in this case. It seems to me that the concepts are very complex so that we need at least some nonlinearity on the predictive function? Or it is a biased estimation which may cause other theoretical problems such as identifiability of the missing concepts.

3. Please justify your method more comprehensively in the experiment section. Say more experiments or some structure analysis of your model. The result-reproducing possibility is also not good for this version.

---

### Official Review · Reviewer_nngX · 2024-11-03

**Soundness:** 2
**Presentation:** 2
**Contribution:** 2
**Rating:** 3
**Confidence:** 5

**Summary:**

The paper tries to address the problem of estimating the causal effect in terms of an intervention in terms of  human interpretable concepts that represent high level knowledge. Existing methods all require that the concept set is completely annotated and known in advance however this paper tries to tackle the situation where some of these may be missing. The authors show that having missing concepts can bias the estimation of the treatment effect and introduce a means of overcoming this through using a Missing Causal Concept Explainer. The authors claim the approach enables them to account for residual bias from missing concepts and allows them to achieve promising performance based on validation with a real data set.

**Strengths:**

The idea of using concepts to interpret the treatment effect of an intervention is a really nice one: just as doctors dont recall precise details of earlier patients when determining the course of treatment for a new patient, but rather perform diagnostics on the basis of some high level characteristics that patients may share, this approach seems like the right way to think about how to view effects of interventions in practice.

**Weaknesses:**

1) The paper is missing references to crucial pieces of work that have previously explored treatment effect estimation using concepts before (see this work by Goyal et al 2019, https://arxiv.org/abs/1907.07165). As such the novelty of the work is limited. While i understand the contribution may be viewed as the fact that this work addresses the issue of missingness of concepts while prior work does not, I feel that in order to address missingness adequately, the work is missing metrics that might help quantify concept completeness for instance (see work by https://openreview.net/forum?id=tglniD_fn9).

2) Crucially, if the work is indeed about inferring the treatment effect of an intervention in the presence of missing concepts, this might be framed as the problem of learning the treatment effect of an intervention where some concepts are known and some concepts are unknown. When concepts are unknown, there may be certain biases or shortcuts that indirectly impact outcomes, whose effects you would like to mitigate. This problem has been extensively studied in several papers on shortcut learning. (see for instance https://proceedings.neurips.cc/paper_files/paper/2022/file/d791394d32c428aecc7a5b101fb47799-Paper-Conference.pdf). Unfortunately without mentioning this work and prior works on shortcut learning, it seems the authors have overlooked major works that bare close resemblance to what is done here (with the exception of using this on text data)

3) There is only one data set containing textual information that is used to validate the approach. This makes it very difficult to assess what the performance of the method looks like (especially where other metrics such as concept completeness or similar are not quantified and other important baselines are missing)

4) The authors need to make the assumptions for this paper very clear. What is meant by causal here? What do you intervene on in order to estimate the causal effect? How do these interventions impact the distributions of concepts learnt and inferred.  What is the SCM associated with this concept-based predictor or underlying causal structure that is assumed throughout the paper? In for instance, the CACE paper (https://arxiv.org/pdf/1907.07165), the authors show a clear graph of the assumed causal structure in Fig 1. Similarly in the work on shortcuts (https://proceedings.neurips.cc/paper_files/paper/2022/file/d791394d32c428aecc7a5b101fb47799-Paper-Conference.pdf), we have the causal structure shown in Fig 1.

5) The work explicitly focuses on computing the individual causal concept effect, but concepts might be correlated or share certain features in which case intervening on one concept should also impact the others, depending on how they are correlated. How does your approach account for this?

6) I would like to see some qualitative and quantitative results on the number of times the approach is able to detect and overcome bias in the data and what it detects as the bias in the first place. In the work on shortcuts I mentioned earlier, the authors mitigate the effects of a shortcut on the outcome by regularizing the model accordingly to downweight its contribution; however, this approach only works under specific conditions and assumptions and can sometimes overconstrain the model/end up introducing additional sources of bias. Do you observe similar such behaviour with your approach? If not, why not? Can we view the key contributors to predictive performance and why these make sense?

**Questions:**

1) Can you distinguish your work from work by Goyal et al 2019, (https://arxiv.org/abs/1907.07165) and especially Wang et al 2023, (https://proceedings.neurips.cc/paper_files/paper/2022/file/d791394d32c428aecc7a5b101fb47799-Paper-Conference.pdf)

2) Can you make the assumptions behind your approach very clear in a section explicitly marked assumptions and state the conditions under which these would be plausible/why they make sense. Also include a figure showing the assumed causal structure.

3) How do you deal with the fact that concepts may be correlated when computing the ICACE?

4) Can you demonstrate the method's performance on more datasets other than text data. I would also like to see more results on the number of times you are able to detect biases and correct them, how you downweight the contribution of these biases/integrate them out. (see point 6) above)

5) How does performance get impacted if there are more than one sources of bias?

6) What happens to performance as more concepts are missing proportional to the information available?

---

### Official Review · Reviewer_ruUy · 2024-11-05

**Soundness:** 2
**Presentation:** 1
**Contribution:** 2
**Rating:** 3
**Confidence:** 4

**Summary:**

The paper address estimation of causal effect of concepts on a model.
They argue that the estimation is biased due to unobserved concepts, which is common in many applications.
A missingness-aware concept explainer is proposed that decomposes the encoded representation to contain contributions from observed and unobserved concepts.
The contribution of observed concepts is assumed to be orthogonal to unobserved ones, which they argue enables unbiased estimation.

I do not think the paper meets the quality standard of the ICLR conference. I had trouble understanding their motivation, proposal, and discussion of results.

**Strengths:**

The paper addresses a practical problem of estimating the causal effect of certain concepts with incomplete concept set.

**Weaknesses:**

**Model assumptions**.
The paper did not argue well all the model assumptions. (1) Why are unobserved concepts assumed to be orthogonal to the observed ones?
I believe the estimation is far more tricky when the observed/unobserved concepts are correlated.
(2) Why is the concept-to-output mapping assumed linear? Many people make the linearity assumption, but rationale is needed here nevertheless.
(3) L 228-229 "We hypothesize that H contains all necessary information about all concepts, including unobserved ones." What does unobserved concepts really mean?
Can we talk about everything unobserved as one big concept? In which case, $\hat{\beta}_{un}$ is just a scalar, i.e. one-dimensional?

**Poor empirical evaluation**.
The paper only evaluates on one dataset called CEBaB, and ends the paper with the statement (that I don't agree): "Further validation will be available only when more benchmark datasets for causal concept effect estimation are available."
I believe the paper could be made far stronger by evaluating on synthetic datasets that demonstrate how their method enabled unbiased estimation.

From Table 1, I observe that MCCE is not statistically significant over CPM. Since, their method is simple, I expected stronger empirical validation.
I do not also understand the need for many different metrics: L2, Cosine, Norm.

Table 2 argues that MCCE's performance is robust even when some concepts are unobserved. But we need to also see vanilla fitting using $N(X)\approx C_{ob, X}^T\beta_{ob}$ for comparison.
If using only observed concepts is decreasing the performance by a lot, that is when the point of Table 2 really comes out.

**Questions:**

Please see the Weakness Section.

---

### Official Review · Reviewer_mkrQ · 2024-11-07

**Soundness:** 1
**Presentation:** 2
**Contribution:** 2
**Rating:** 3
**Confidence:** 3

**Summary:**

This paper introduces MCCE (Missingness-aware Causal Concept Explainer), a framework for estimating causal effects of concepts on machine learning model behavior when some concepts are unobserved. The key innovation is the construction of pseudo-concepts that are orthogonal to observed concepts, which helps capture information from unobserved concepts and reduces bias in causal effect estimation, all while using a linear predictor without requiring counterfactual data. Validated on the CEBaB dataset with three different language models, MCCE achieves comparable or superior performance to existing methods while providing both local and global interpretations, though its empirical validation is limited to a single text-based dataset despite claiming modality-agnosticism.

**Strengths:**

1.This paper addresses an important and previously unexplored gap in concept-based explanations: handling unobserved concepts.

2. This paper is clearly written and easy to follow.

**Weaknesses:**

1. This paper makes a very strong assumption: $H$ contains all necessary about all outputs.

2. This paper does not prove that $\hat{\beta} _{ob}$ estimated by the proposed MCCE is an unbiased estimation of $\beta _{ob}^*$.

2. The number of pseudo-concepts ($j$) is a critical hyper-parameter, there's no theoretical justification for choosing this number.

3. This paper only conduct experiments on one test dataset. The authors should at least do experiments on another image dataset.

**Questions:**

N/A

---

### Note · Authors · 2024-11-13

I have read and agree with the venue's withdrawal policy on behalf of myself and my co-authors.